# From Random Perturbation to Precise Targeting: A Comprehensive Review of Methods for Studying Gene Function in *Monascus* Species

**DOI:** 10.3390/jof10120892

**Published:** 2024-12-23

**Authors:** Yunxia Gong, Shengfa Li, Deqing Zhao, Xi Yuan, Yin Zhou, Fusheng Chen, Yanchun Shao

**Affiliations:** 1College of Food Science and Technology, Wuhan Business University, Wuhan 430056, China; gongyunxia@wbu.edu.cn (Y.G.); yuanxiraiden@163.com (X.Y.); yinzhou@wbu.edu.cn (Y.Z.); 2College of Food Science and Technology, Huazhong Agricultural University, Wuhan 430070, China; li33hjhj@163.com (S.L.); delaneyzhao@163.com (D.Z.); 3Hubei International Scientific and Technological Cooperation Base of Traditional Fermented Foods, Huazhong Agricultural University, Wuhan 430070, China

**Keywords:** *Monascus* spp., genetic transformation, site-specific gene editing, multiple-gene perturbations

## Abstract

*Monascus*, a genus of fungi known for its fermentation capability and production of bioactive compounds, such as *Monascus* azaphilone pigments and Monacolin K, have received considerable attention because of their potential in biotechnological applications. Understanding the genetic basis of these metabolic pathways is crucial for optimizing the fermentation and enhancing the yield and quality of these products. However, *Monascus* spp. are not model fungi, and knowledge of their genetics is limited, which is a great challenge in understanding physiological and biochemical phenomena at the genetic level. Since the first application of particle bombardment to explore gene function, it has become feasible to link the phenotypic variation and genomic information on *Monascus* strains. In recent decades, accurate gene editing assisted by genomic information has provided a solution to analyze the functions of genes involved in the metabolism and development of *Monascus* spp. at the molecular level. This review summarizes most of the genetic manipulation tools used in *Monascus* spp. and emphasizes *Agrobacterium tumefaciens*-mediated transformation and nuclease-guided gene editing, providing comprehensive references for scholars to select suitable genetic manipulation tools to investigate the functions of genes of interest in *Monascus* spp.

## 1. Introduction

*Monascus* spp. are filamentous fungi, which are heterotrophic eukaryotic micro-organisms with hairy hyphae and sophisticated organelles (Figure 1a) [1]. Species delimitation of *Monascus* spp. is based on phenotypic characteristics (Figure 1a). This genus can reproduce either vegetatively by forming filaments and conidia or sexually by forming ascospores [2]. Given the rapid development and low cost of sequencing to date, species can be delimited by the genotype, including ITS, partial LSU, and β-tubulin gene sequencing [3]. *Monascus* has often been positioned outside the order *Eurotiales* but in *Aspergillaceae*, and this genus has more than 20 species [2]. In addition, *Monascus* holds economic importance in several areas, such as coloration [4], brewage [5], and biomanufacturing [6] (Figure 1b). At present, modern pharmacological studies have proven that monacolin K (MK) synthesized by *Monascus* spp. can exert a good lipid-lowering effect [7]. Moreover, the various metabolites of *Monascus* spp., such as *Monascus* azaphilone pigments (MonAzPs) and γ-aminobutyric acid (GABA), have been identified to have multiple physiological activities. One of the limitations of the application of *Monascus* spp. is that some *Monascus* strains can synthesize citrinin (CIT), a nephrotoxic fungal toxin [8]. At present, the commercial application of *Monascus* spp. is hindered because of certain challenges, such as controlling harmful substances and shortening the fermentation cycle of active ingredients. Therefore, researchers have focused on the commercialization of *Monascus* spp. for decades.

Optimizing the cultural conditions can effectively shorten the fermentation cycle [9,10]. However, the production of harmful metabolites is uncontrollable. Moreover, under optimized conditions, the production of secondary metabolites (SMs) will gradually stabilize with the metabolic inhibition and restriction of strains, which requires strain improvement. Classical genetics aiming at random perturbation have been applied to strain improvement in the early stage. In general, mutagenesis can artificially increase the number of mutations and expand genetic variability; therefore, these mutants were screened for growth on synthetic media, and the mutants of interest were selected by detecting the target SMs [11]. The wild-type strain of *Monascus* spp. with multiple nuclei (Figure 1a) underwent mutagenesis to construct a random mutation library, which needs considerable work to screen mutants. These obtained mutants were derived from the multiplication of a single haploid cell containing a mutant gene. In order to identify mutant gene, genome sequencing was required to determine the functions of genes associated with the traits of mutants. In the last century, the sequencing technique showed high cost and low precision, which limited the development of mechanistic research in *Monascus* spp. Thus, the recipient genome is intervened via genetic manipulation to construct a controllable mutation library, and lower sequencing quantity is required to reveal the function of genes associated with various phenotypes. Naturally, transformation techniques have been developed to promote the penetration of targeted exogenous DNA to cell barriers. Among these techniques, genetic manipulation, which is suitable for *Monascus* spp., has undergone random perturbation and precise editing. Thus, this review summarizes most of the genetic manipulation tools used in *Monascus* spp., particularly *Agrobacterium tumefaciens*-mediated transformation (ATMT) and nuclease-guided gene editing, thereby providing comprehensive references for scholars to compare and select suitable genetic manipulation tools.

## 2. Genetic Transformation Promotes the Construction of a Controllable Mutation Library

Genetic transformation, which promotes the penetration of exogenous DNA to cell barriers, is necessary for reverse genetics research. Therefore, chemical, physical, or biological methods have been developed to efficiently identify gene functions (Figure 2). The main chemical method includes preparation of high-quality competent cells using different reagents, such as protoplast-mediated transformation (PMT) and restriction enzyme-mediated integration (REMI) based on PMT. The primary physical method consists of applying a high-strength electric field to damage cell walls instantly, such as biolistic transformation (BT), electroporation (EP), and high-density distributed electrode network (HDEN). PMT and ATMT have been widely applied in biological transformation, greatly promoting the study of gene functions.

### 2.1. Chemical and Physical Genetic Transformation Techniques

In general, cells that transform into a competent state are the preferred recipients for genetic transformation [12]. Genetic transformation in *Monascus* spp. involves increasing the permeability of the cell wall via enzymatic hydrolysis and promoting the absorption of exogenous genetic materials. Therefore, PMT is a widely used genetic transformation technique. At present, cellulase, snails, and lysing enzymes have been proven effective in removing cell walls [13,14,15]. In addition, the supplementation of osmotic pressure stabilizers, such as NaCl, MgSO_4_, sorbitol, and mannitol, can protect protoplasts from water swelling [16]. Normally, inorganic salts promote the release of protoplasts, and sorbitol and mannitol can increase the regeneration rate of protoplasts [17]. However, the preparation of protoplasts varies among different species [18], which should be adjusted in accordance with the type of *Monascus* spp. As an application extension of PMT, REMI has been developed to construct mutation libraries [19], which include two stages (Appendix A): linear DNA with sticky ends generated by RE digestion and integration of linearized DNA into the recipient [20]. Once the mutants were created, the genome was then digested into a variety of fragments and ligated using a vector (also known as plasmid rescue) for sequencing. Therefore, linking mutant phenotypes with gene function is an important transformation method, which allows random perturbation in the genome of *Monascus* spp. However, the application of different REs would lead to different numbers of mutants; therefore, screening suitable REs is helpful for the construction of a random mutation library. However, the selection of REs is a time-consuming process, which limits the wide application of the REMI technique.

Physical transformation is based on the application of a moderate external force, including BT, EP, and HDEN. In general, exogenous DNA carrying a resistance gene is transferred into the receptors, such as the conidia, hyphal fragments, and protoplast. However, these external forces can cause great damage. For the BT test, only two hygromycin-B-resistant colonies were obtained from 0.5 cm^3^ (about 0.2 g) of wet conidial masses for *M. purpureus* [21]; regarding the EP, six mutants were obtained from 10^9^ protoplasts using *M. purpureus* as the parent [22]. Therefore, a method for the precise control of pulse parameters was developed: HDEN. As an application in *Monascus* spp., a uniform electric field was generated in the spore solution to directly introduce exogenous genetic materials into the dormant spores of *Monascus* spp. [23]. Spores were the preferred receptors, especially germinating spores, not dormant spores [24]. Overall, these physical methods are efficient, but their destructive effects on recipient cells result in low genetic transformation rates, and certain devices limit their application.

### 2.2. ATMT: A Type of Biological Genetic Transformation Technique

ATMT mainly includes the following steps (Figure 2): (1) preparation of conidia, (2) inoculation and incubation of *A. tumefaciens* containing the plasmid with the T-DNA region, (3) co-culturing of conidia and *A. tumefaciens* on filters, (4) transfer of filters to a selection medium for incubation, (5) screening of the transformant. This gentle process enables ATMT to exhibit high transformation efficiency, and the T-DNA region can be randomly integrated into the recipient to produce numerous mutated phenotypes, which are easily linked to responsible genes [25]. Before the completion of a large number of genome sequencing, the ATMT technique was performed in *Monascus* spp. to construct a mutation library. Using *M. ruber* M7 as the parent, a mutation library of T-DNA insertion, including 5132 transformants, was constructed, and the mutant strains of interest were further screened [26]. Subsequently, thermal asymmetric interlaced PCR was performed to amplify the DNA sequences that flanked the T-DNA of the mutant strains of interest. Then, these fragments were sequenced and analyzed using the Blast tool [27]. As an application of ATMT into *M. purpureus*, T-DNA random mutagenesis was utilized to successfully localize the genes responsible for the biosynthesis of MonAzPs [28], which harbored a T-DNA insertion upstream of a transcriptional regulator gene (*mppR1*), underscoring the importance of this technique in uncovering complex biosynthetic pathways. Through phenotypic analysis and sequencing, the phenotypes of *Monascus* spp. are associated with genes, which greatly promotes gene function research and genetic breeding.

## 3. Site-Specific Gene Editing Facilitates the Genetic Resource Mining of *Monascus* spp.

Given the advances in genome sequencing and algorithms in bioinformatics, large-scale genome data help researchers manipulate and activate microbial resources [29], thereby promoting efficient analysis of numerous unknown genes. Compared with random perturbation, achieving site-specific gene editing by inducing artificially designed exogenous DNA carrying screening markers into *Monascus* spp. is an effective scheme. Therefore, ATMT has been developed for site-specific gene editing. In particular, the application of ATMT depends on the modified binary-vector system with the T-DNA region (Figure 3a). For site-specific gene editing, the left and right border sequences of exogenous DNA (also known as the T-DNA region) were amplified from the homology arms (HAs) of the target region (Figure 3b). In the last decade, our research group has completed the whole-genome sequencing of many *Monascus* strains. Therefore, in order to analyze the gene function of interest, we first applied the bioinformatics tool to analyze sequence information and then designed the T-DNA region (cassette) carrying the homologous sequences on both sides of the target sequence and antibiotic resistance genes (ARGs). Site-specific gene editing strains can be obtained by applying modified ATMT and Southern blot verification. Followed by physiological and biochemical analyses, the function of the target gene can be announced [30]. In some cases, the deletion of particular genes is fatal to *Monascus* spp.; thus, RNA interference provides the strategies to deal with this dilemma at the post-transcriptional level through double-stranded RNA. The actuation of RNAi can be effectively prolonged by transforming the exogenous genetic material in ATMT, which greatly improves the inhibition efficiency of target genes. Our team introduced RNAi into *M. ruber* M7, and intron-containing hpRNA interference was generated to silence the transcripts of *pksCTα/β* responsible for CIT biosynthesis, which hindered the citrate cycle (TCA cycle) and decreased the production of biosynthetic precursors. The targeted mutant was constructed with a 90% decrease in CIT yield [31].

Targeted gene manipulation in *Monascus* spp. addresses the polynuclear and heteronuclear phenomena and deals with mitotically unstable and high-frequency homologous end joining (NHEJ). Thus, the perturbation of the NHEJ pathway is an effective way to improve gene replacement frequencies (GRFs). For example, our previous research adopted a targeted approach to enhance the gene targeting efficiency in *M. ruber* M7 by perturbing the NHEJ pathway. The GRF of the knockout genes *ku70*, *ku80*, and *lig4* of *M. ruber* M7 reached 42%, 62%, and 85%, respectively, which resulted in a two-, three-, and four-fold increase in GRF compared with the wild type [32,33]. This enhancement was demonstrated by targeting the *ku70* and *triA* genes, thereby achieving GRFs of 68% and 85%, respectively. The screening of the genetic transformant relies on ARGs into the genome of the recipient. In addition, the genes responsible for biosynthesis of SMs are clustered on the genome, which is known as the biosynthetic gene cluster (BGC) [34,35]. Therefore, research on the function of genes in BGC has been hindered because of the limitation of ARGs suitable for *Monascus* spp., and these high-efficiency mutants cannot be used for multi-gene modification. Thus, nutritional screening has become the preferred option to address this dilemma, particularly the “Latour system” (Appendix A). In general, constructing nutrient-deficient strains is difficult; however, a *pyrG*-deficient strain has been successfully constructed using *M. ruber* M7 as the parent after several years of attempts. Then, the endogenous gene *mrpyrG* instead of ARGs was developed as a screening marker, and multi-gene knockout mutants, including Δ*pigG*, Δ*pigI*, Δ*pigG*Δ*pigH*, and Δ*pigG*Δ*pigH*Δ*pigI*, were constructed using Δ*pyrG*Δ*lig4* as the parent, resulting in an 18-fold increase in GRF [36]. After further analysis of the phenotype and metabolism, Uridine addition has been found to influence the production of MonAzPs. In an unpublished study, an overexpression mutant that can encode the transcription factor *mokH* in MK BGC was constructed using *pyrG* as the screening marker; however, the MK yield of this mutant was significantly lower than that of *M. pilosus* MS-1, an industrial strain with high MK production. These results indicated that the application of the “Latour system” in *Monascus* spp. was limited to commercial applications. Hence, establishing an efficient and precise gene editing technology that can disturb multiple genes simultaneously in the postgenomic era is necessary.

## 4. Precise Gene Editing by Sequence-Specific Nucleases

For higher gene editing efficiency, sequence-specific nucleases (SSNs), which consist of a DNA recognition domain and a nuclease domain (Appendix A), have become a research hotspot worldwide. The application of SSNs for large-scale precise gene editing is prompted by the development of the CRISPR/Cas system, which has revolutionized genetic engineering, providing unprecedented precision and efficiency in gene editing.

### 4.1. CRISPR/Cas System Has Been Developed as an Effective Gene Editing Tool

The CRISPR/Cas system has been proven effective in promoting acquired immunity in bacteria to resist the invasion of exogenous genetic materials. As shown in Figure 4a, this defense system works according to the stages of adaptation and interference [37]. The CRISPR/Cas system can be classified into two major categories (Class 1 and Class 2) and six types (types I–VI) depending on the active form of the Cas protein [38]. At present, Class 2 CRISPR/Cas system has been developed as a third-generation SSN technology, and the CRISPR/Cas9 system is still the commonly used system. As a gene editing system, the working process of Class 2 CRISPR/Cas system can be briefly described as follows: crRNA (including the protospacer) paired with tracrRNA has been artificially modified into chimeric sgRNA [39], and the Cas9 endonuclease is activated by recognizing the PAM sequence to cleave the target locus and produce DSBs [40].

### 4.2. Application of the CRISPR/Cas9 System to Gene Editing for Monascus *spp*.

The production of DSB relies on Cas proteins exerting cleavage activity, and a single Cas protein can simplify the process and increase applicability compared with Cas protein complexes; thus, the CRISPR/Cas9 system is the preferred strategy. In regulating the biosynthesis of CIT, Cas9, which constitutively expressed a chassis strain, was constructed by ATMT using *M. purpureus* as the parent, and then, variable sgRNAs were introduced by PMT to obtain *pyrG* and *ctnD* double gene-edited strains, leading to more than 91% and 98% reduction in CIT levels for mycelium and fermented broth, respectively [41]. For a wide application of the CRISPR/Cas9 system in filamentous fungi, Nodvig et al. designed a versatile CRISPR/Cas9 system by modifying the Cas protein and sgRNA to actualize simplification and flexibility (Figure 4b) [42]. In particular, the optimized Cas9 from *S. pyogenes* and the 3′-extended sequence encoding a SV40 nuclear localization signal (PKKKRKV) were inserted into the *tef1* promoter and terminator (P*tef1*-T*tef1*) of *A. nidulans*. In addition, sgRNA released by the 5′-end hammerhead and 3′-end hepatitis delta virus in the nucleus was liberated from a larger transcript controlled by the gpdA promoter (P*gpdA*) and *trpC* terminator (T*trpC*) of *A. nidulans*. Moreover, these components, in addition to ARGs, were inserted downstream of the AMA1 sequence for independent replication.

To carry out precise gene editing, researchers have attempted to apply this versatile CRISPR/Cas9 system into *Monascus* spp. As the first test in *Monascus* spp., this versatile CRISPR/Cas9 system has been applied to *M. purpureus* to knock out the 15-kb CIT BGC, resulting in 2–5% increases in MonAzPs production [43]. Similarly, the mutants of *pigF* encoding oxidoreductase were constructed to increase pigment purity in *M. purpureus*, and two negative regulator genes *pigI*/*pigI′* were inactivated to overproduce MonAzPs in *M. ruber* [44]. In applying this versatile system in *Monascus* spp. to achieve gene manipulation, completing highly efficient PMT in *Monascus* spp. is of great importance. More than 10 species belonging to the genus *Monascus* are recognized internationally, and they have interspecific diversity with different characteristics; therefore, the implementation of PMT must be adjusted adaptively. In our research, highly active protoplasts were first obtained via enzymatic hydrolysis. Then, this versatile system was successfully introduced into *M. pilosus* MS-1 for gene editing [18]. Using this operation, an improved mutant was obtained with enhanced MK yields, which exhibited excellent commercial application potential.

### 4.3. Precise Base Editing

In general, the genome of filamentous fungi is several tens of megabytes, and the 20 nt recognition sequences of sgRNA might lead to off-target effects. Researchers found that the mutants of the Cas9 protein can perform precise substitution of a single base. Li et al. developed a base editing (BE) system [45], including nCas9 and cytidine deaminase. The D10A mutation of Cas9 leads to the production of single-stranded DNA breaks, and cytidine deaminase converts cytosine (C) to uracil (U) with the addition of a uracil glycosylase inhibitor for maximum efficiency. Given its simplicity and efficiency [46], the CRISPR-BE system has achieved eight-gene disruption in *A. nidulans* with GRF of 40%; however, this system has not been applied to *Monascus* spp. Instead, a helicase-guided BE system, which consists of a cohesin–dockerin complex and a conserved DNA helicase protein, has been developed, which can facilitate the self-assembly of the DNA helicase protein with diverse BEs. The DNA helicase protein could unwind the double-stranded DNA and break the hydrogen bonds between the two DNA strands. Thus, transient single-stranded DNA was allowed to introduce edited bases at random loci throughout the chromosome. This helicase-guided BE system was applied to *M. purpureus* to generate improved strains with enhanced MK yields, and the removal of plasmid in mutants was achieved via passaging for several generations. For precise gene editing, a marker-free strain with a 10-fold increase in MK production was obtained, which demonstrated excellent commercial value.

## 5. Conclusions and Perspectives

With the advent of the postgenomic era, the combination of bioinformatics and gene editing has made it easier to obtain filamentous fungal transformants that produce less harmful substances or higher value-added biological products. Integrating genetic manipulation techniques with comprehensive metabolic pathway analysis could provide insights into the regulatory mechanisms underlying SM biosynthesis, such as MK and MonAzPs. Various optional molecular manipulation tools can provide convenience in exploring the gene function of *Monascus* spp. to enhance metabolite production and environmental adaptability (listed in Appendix A). In addition, exploring the potential application of *Monascus* species in producing novel functional compounds and developing high-functionality foods should be a priority. Regardless of directly changing the DNA sequence or transcriptional interference, different genetic transformation methods and gene manipulation techniques have provided researchers with pathways to seek the biological resources of *Monascus* spp. Otherwise, leveraging advancements in machine learning and artificial intelligence could further enhance the predictive capabilities and streamline the development of safe and sustainable *Monascus*-derived products. The research of comparative studies across different *Monascus* species to identify commonalities and differences in metabolic pathways and genetic regulation will provide a more comprehensive understanding of the potential application of these species in biotechnology and medicine. In achieving higher gene editing efficiency, SSNs, which consist of a DNA recognition domain and a nuclease domain (Appendix A), have sparked a research boom worldwide. Furthermore, the application of SSNs for large-scale precise gene editing is brought about by the development of the CRISPR/Cas system, which has revolutionized genetic engineering, providing unprecedented precision and efficiency in gene editing.

## Figures and Tables

**Figure 1 jof-10-00892-f001:**
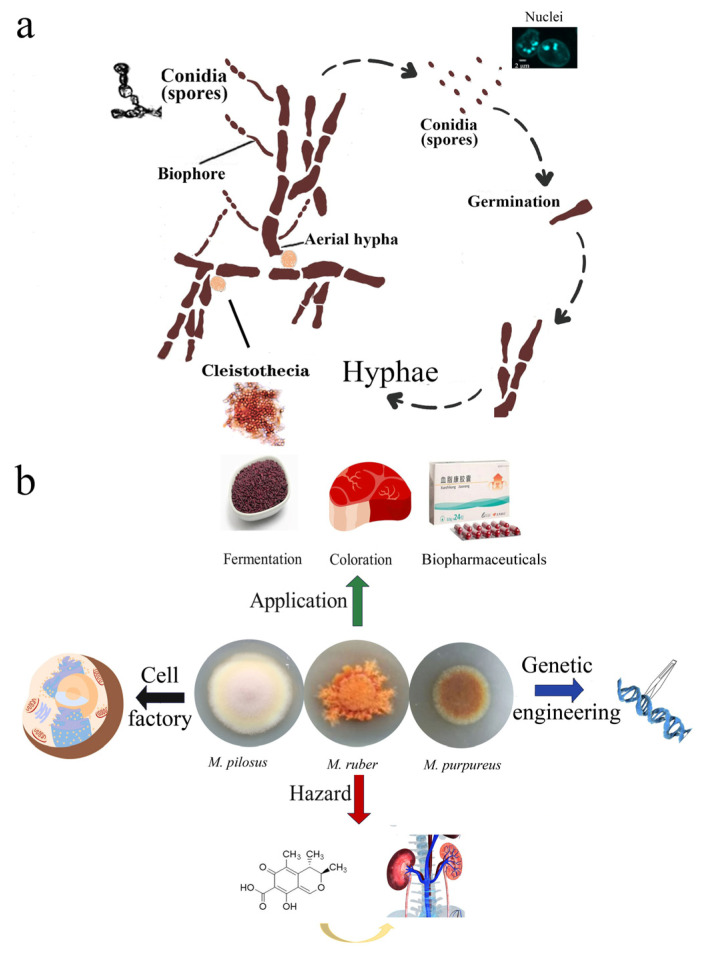
Physiology and application of *Monascus* spp. (**a**) Physiology of *Monascus ruber* M7. (**b**) Application and hazard of *Monascus* spp.

**Figure 2 jof-10-00892-f002:**
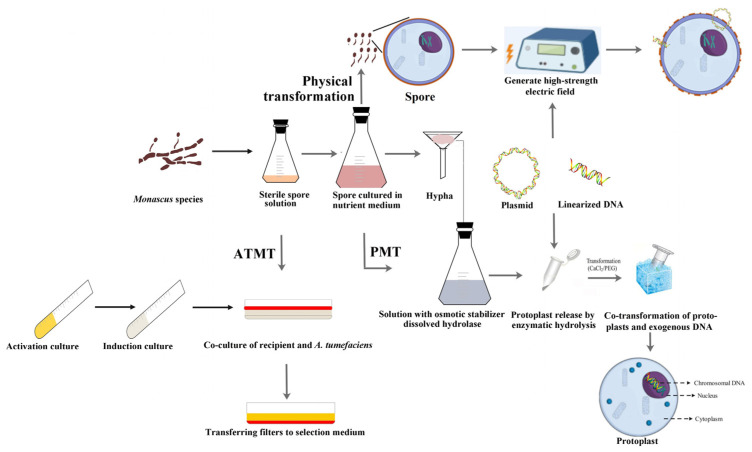
Genetic transformation applied in *Monascus* spp. The chemical transformation technique refers to PMT; the physical transformation technique refers to imposition of an external force; and the biological transformation technique refers to ATMT. These techniques have been developed to promote the penetration of exogenous DNA to cell barriers.

**Figure 3 jof-10-00892-f003:**
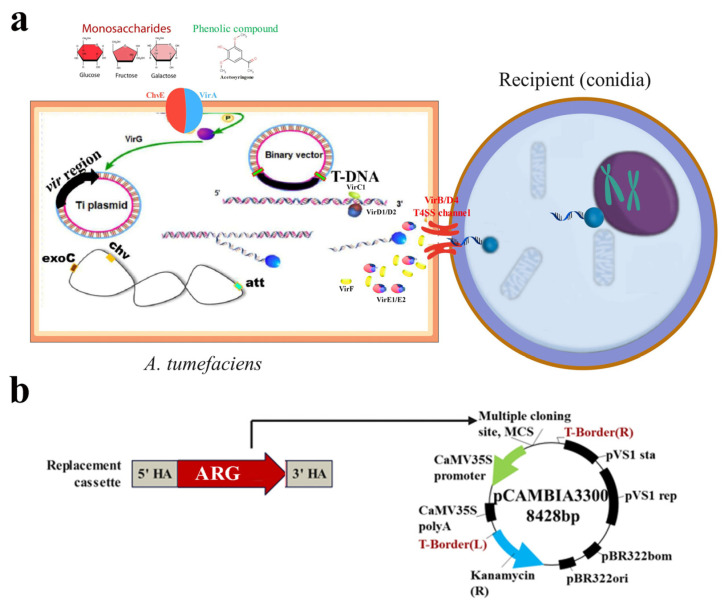
Application of ATMT to gene editing. (**a**) Principle of ATMT for gene editing. (**b**) Map of the artificially designed plasmid for constructing site-specific gene editing strains in our lab.

**Figure 4 jof-10-00892-f004:**
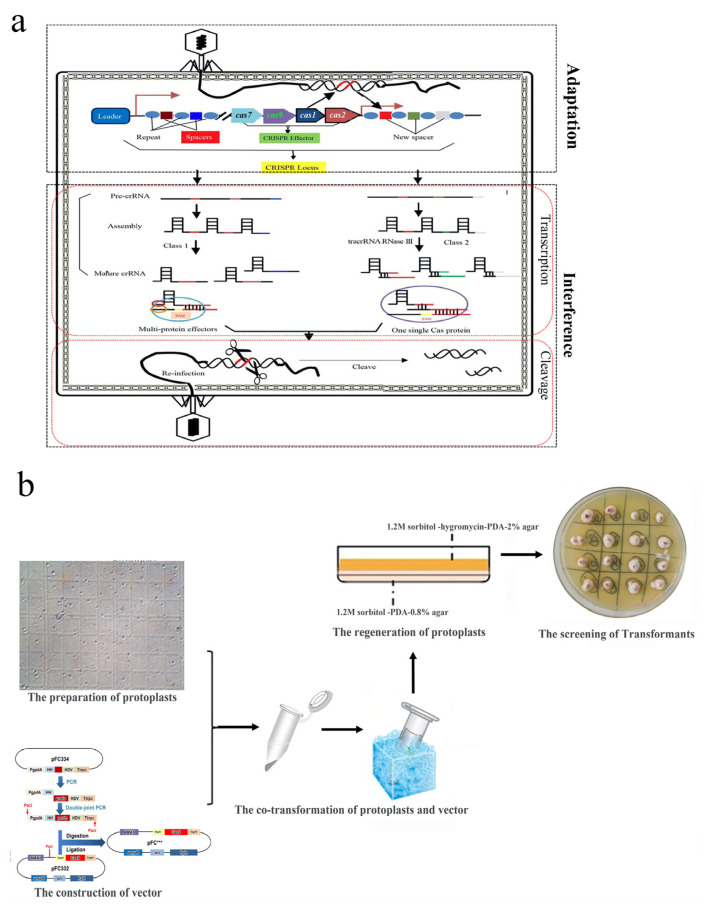
Principle and application of the CRISPR/Cas gene editing system. (**a**) Principle of the CRISPR/Cas gene editing system, including Class 1 and Class 2 CRISPR/Cas systems. (**b**) Application of the CRISPR/Cas9 gene editing system in *Monascus* sp.

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
