# Peer review of "From Random Perturbation to Precise Targeting: A Comprehensive Review of Methods for Studying Gene Function in Monascus Species"

_jof, 2024, doi:10.3390/jof10120892_

Round 1
Reviewer 1 Report
The article is very clear and nicely written, but I miss a summary, in the form of a table, of the successes achieved in suppressing citrinin production and increasing pigment or monacolin production by the various methods described here.
I have no other comments.
Author Response
Comment 1: The article is very clear and nicely written, but I miss a summary, in the form of a table, of the successes achieved in suppressing citrinin production and increasing pigment or monacolin production by the various methods described here.
Response: Thank you for your kind suggestion. A table including a summary of various methods applied into Monascus spp. has been added into the supplementary files.

Reviewer 2 Report
The review is titled "A comprehensive review of methods to study the gene function in Monascus species", however the review largely describes well-known modern methods for studying genetic engineering of fungi, and pays relatively little attention to the specificity of their application to Monascus species. The authors need to significantly revise the review, exclude numerous data related to the generally accepted tools of genetic engineering, and focus on a detailed description of the direct manipulations carried out with Monascus species. In addition, the title of the manuscript does not correspond with the material presented. The authors state that the topic will be covered: "From random perturbation to precise targeting: A comprehensive review of methods to study the gene function in Monascus species". However, it is not clear what "random perturbation" is, maybe it is random mutagenesis for strain improvement. In any case, It is necessary to describe what work on "random perturbation" has been done for Monascus species previously and to cite the relevant work or change the title of the manuscript.
the authors of the review talk in great detail about chemical and physical transformation methods (about osmoprotectors such as NaCl, MgSO4, sorbitol mannitol, the "gene gun" method, gold and other carriers for electroporation), Lines 77-104. Much of the knowledge presented is in textbooks and well known to the public. It is unclear why the latest knowledge on Monascus species should be diluted by this.
The section on the biological transformation method (Lines 115-143) again presents common knowledge that is more appropriate for a textbook than for a review of a specific group of organisms (about the binary-vector system, Ti plasmid, vir genes). In this section, Lines 143-162 are devoted to work with Monascus species. However, then, in Lines 166-186 (except Lines 170-171, where there is a reference to Monascus), banal truths and known systems are listed. For example, the authors state: "In addition, the genes responsible for biosynthesizing secondary metabolites (SMs) are clustered on genome, called biosynthesis gene cluster (BGC)". Or the authors describe a screening system in Saccharomyces cerevisiae based on 5-fluoroorotic acid (5-FOA). However, it does not need to be described in such detail in a review devoted not to selection systems as such. It is enough to describe how this selection system was successfully applied to the construction of M. ruber M7. And then only 187-200 again about Monascus
Speaking about the CRISPR/Cas system, the authors make a literature review, starting from 1987, Lines 209-247 - representing educational material (where the bacterial immunity system and many other fundamental knowledge are described) not related to Monascus. For example, the phrase "When the bacteriophage invades the bacteria, the featured segment of exogenous DNA (also called protospacer) will be recognized and then integrated into the CRISPR array to create a new spacer, thus completing the stage of adaptation" And only in section 247-281 the knowledge is given, related to Monascus.
4.1. The principle of RNAi
It is unclear why the review provides a theory on the principle of RNAi 309-334. It is enough to start with the work of Romano et al. Line 340-342 and further, where the authors paid attention to the peculiarities of the application of RNAi for fungal strains.
Comments
Figure 1
Drop by signature size. In its current form, the figure is poorly readable, some signatures are illegible, others are too bulky and clutter the arrows
Figure 2, 3
Rework Figure 2, 3, placing A and B not horizontally, but vertically, so that it is easier to read in the book version of the page orientation in the manuscript
Fig. 2B – in the figure under the fungus icon it is written - filamentous fungus, specify the name of the fungus since in the caption to the Figure it is indicated – Agrobacterium tumefaciens-mediated transformation in Monascus spp.
Figure 4.
This Figure is taken directly from the article "The RNAi Universe in Fungi: A Varied Landscape of Small RNAs and Biological Functions", doi: 10.1146/annurev-micro-090816-093352 (Figure 1 in the original 2017 article). The authors of the current review have changed the design of the original Figure (e.g., two RNA strands), but not its content, cut off the bottom arrows in the original Figure, and did not cite the original source in the Figure legend, which needs to be corrected. In addition, instead of an expanded legend to this figure in the original source, where the name and function of the components of the systems are described, the authors wrote "The principle of RNA interference in Monascus spp." However, the left scheme in the original Figure is given for Neurospora crassa, and the right part in the original Figure is given for Mucor circinelloides. Authors must cite experimental work confirming that these mechanisms are also involved in Monascus spp in order to include this borrowed diagram in their review. Otherwise, this figure should not be included in a review of Monascus spp., since it is not related to the experimental study of these organisms, but is a template from other fungi.
Author Response
Major comments
Comments 1: The review is titled "A comprehensive review of methods to study the gene function in Monascus species", however the review largely describes well-known modern methods for studying genetic engineering of fungi, and pays relatively little attention to the specificity of their application to Monascus species. The authors need to significantly revise the review, exclude numerous data related to the generally accepted tools of genetic engineering, and focus on a detailed description of the direct manipulations carried out with Monascus species.
Answer: Thank you for your kind request. The text describing well-known modern methods has been replaced with descriptions of examples of different methods applied in Monascus spp., and an analysis of the advantages and disadvantages of the methods has been added at the end of each section, all chances have been marked in red font, shown on line 104-108, 119-121, 128-135 and 192-195.
Comments 2: In addition, the title of the manuscript does not correspond with the material presented. The authors state that the topic will be covered: "From random perturbation to precise targeting: A comprehensive review of methods to study the gene function in Monascus species". However, it is not clear what "random perturbation" is, maybe it is random mutagenesis for strain improvement. In any case, it is necessary to describe what work on "random perturbation" has been done for Monascus species previously and to cite the relevant work or change the title of the manuscript.
Answer: Thank you for your inquiry. The techniques, such as REMI and BT, described in this review aiming to construct T-DNA mutation libraries are methods of random gene perturbation. The mutants with visible differences were screened from a large number of mutation libraries, then, the DNA sequences were determined by sequencing techniques, and then traits were linked to genes. However, many regulatory genes do not participate in phenotypic changes, so the function of these genes is extremely difficult to be clarified by random perturbation method. With the completion of whole genome sequencing and the release of more gene function annotations, it is an accurate editing method to apply comparative genomics methods into clarify the location and sequence information of target genes, and to achieve site-specific gene editing by performing genetic manipulation.
Detail comments
Comments 3: the authors of the review talk in great detail about chemical and physical transformation methods (about osmoprotectors such as NaCl, MgSO4, sorbitol mannitol, the "gene gun" method, gold and other carriers for electroporation), Lines 77-104. Much of the knowledge presented is in textbooks and well known to the public. It is unclear why the latest knowledge on Monascus species should be diluted by this.
Answer: Thank you for your kind request. The text describing well-known knowledge has been deleted. The CRISPR/Cas9 system applied into gene editing of Monascus strains at this stage relies on PMT technique. Meanwhile, in our case, we also found that different strains require different reagents for protoplast preparation. Therefore, in this manuscript, we cited the factors that affect the protoplast preparation of Monascus spp. at this stage, hoping to provide reference for researchers, shown on Line 91-108.
Comments 4: The section on the biological transformation method (Lines 115-143) again presents common knowledge that is more appropriate for a textbook than for a review of a specific group of organisms (about the binary-vector system, Ti plasmid, vir genes). In this section, Lines 143-162 are devoted to work with Monascus species. However, then, in Lines 166-186 (except Lines 170-171, where there is a reference to Monascus), banal truths and known systems are listed. For example, the authors state: "In addition, the genes responsible for biosynthesizing secondary metabolites (SMs) are clustered on genome, called biosynthesis gene cluster (BGC)". Or the authors describe a screening system in Saccharomyces cerevisiae based on 5-fluoroorotic acid (5-FOA). However, it does not need to be described in such detail in a review devoted not to selection systems as such. It is enough to describe how this selection system was successfully applied to the construction of M. ruber M7. And then only 187-200 again about Monascus
Answer: Thank you for your kind suggestion. The text describing well-known knowledge has been deleted, and the advantages and disadvantages of this system were analyzed, shown on Line 182-195.
Comments 5: Speaking about the CRISPR/Cas system, the authors make a literature review, starting from 1987, Lines 209-247 - representing educational material (where the bacterial immunity system and many other fundamental knowledge are described) not related to Monascus. For example, the phrase "When the bacteriophage invades the bacteria, the featured segment of exogenous DNA (also called protospacer) will be recognized and then integrated into the CRISPR array to create a new spacer, thus completing the stage of adaptation" And only in section 247-281 the knowledge is given, related to Monascus.
Answer: Thank you for your kind suggestion. The text describing well-known knowledge has been deleted, and the fourth part Precise gene editing by sequence-specific nucleases has been modified, which have been marked in red font.
Comments 6: The principle of RNAi
It is unclear why the review provides a theory on the principle of RNAi 309-334. It is enough to start with the work of Romano et al. Line 340-342 and further, where the authors paid attention to the peculiarities of the application of RNAi for fungal strains.
Comments 7: Figure 1
Drop by signature size. In its current form, the figure is poorly readable, some signatures are illegible, others are too bulky and clutter the arrows
Answer: Thank you for your kind suggestion. Figure 1 has been redrawn, and high-resolution images will be submitted.
Figure 2, 3
Rework Figure 2, 3, placing A and B not horizontally, but vertically, so that it is easier to read in the book version of the page orientation in the manuscript
Fig. 2B – in the figure under the fungus icon it is written - filamentous fungus, specify the name of the fungus since in the caption to the Figure it is indicated – Agrobacterium tumefaciens-mediated transformation in Monascus spp.
Answer: Thank you for your kind suggestion. Figure 2 and 3 have been paced in horizontal form, and the description fungus has been changed into Monascus spp..
Figure 4.
This Figure is taken directly from the article "The RNAi Universe in Fungi: A Varied Landscape of Small RNAs and Biological Functions", doi: 10.1146/annurev-micro-090816-093352 (Figure 1 in the original 2017 article). The authors of the current review have changed the design of the original Figure (e.g., two RNA strands), but not its content, cut off the bottom arrows in the original Figure, and did not cite the original source in the Figure legend, which needs to be corrected. In addition, instead of an expanded legend to this figure in the original source, where the name and function of the components of the systems are described, the authors wrote "The principle of RNA interference in Monascus spp." However, the left scheme in the original Figure is given for Neurospora crassa, and the right part in the original Figure is given for Mucor circinelloides. Authors must cite experimental work confirming that these mechanisms are also involved in Monascus spp in order to include this borrowed diagram in their review. Otherwise, this figure should not be included in a review of Monascus spp., since it is not related to the experimental study of these organisms, but is a template from other fungi.
Answer: Thank you for your kind suggestion. Figure 4 is drawn with referring to the article "The RNAi Universe in Fungi: A Varied Landscape of Small RNAs and Biological Functions", and all graphic elements are drawn by us. We are very sorry for your misunderstanding due to our improper quotation. The pictures in the manuscript have been deleted, and the relevant descriptions have been corrected and marked in red font.
Reviewer 3 Report
Yunxia Gong and colleagues present a review article on aspects of transformation and gene editing techniques in Monascus, focusing on Agrobacterium-mediated transformation, CRISPR and RNAi. The article is comprehensibly written and summarizes relevant information for people working with this fungus and beyond. The references cited are appropriate and the figures are informative.
The article suffers in some passages from poor use of language and needs extensive editing, focusing on grammar, syntax and overall correctness and consistency. It is strongly recommended to seek assistance from a native English speaker.
An additional panel highlighting aspects of Monascus physiology (for example a growth test or a microscopy image) would nicely complement Figure 1.
Figures would really benefit from a more detailed description in the relevant figure legends. In this context, parts of the text describing the principle of each method could be accommodated in the relevant figure legend.
The advantages and/or disadvantages of each method could be further highlighted by adding a few more sentences at the end of each chapter or in the conclusions section.
The supplementary figures mentioned in the text were not included in the submission and therefore could not be reviewed. Please make sure to include those in your final version.
Author Response
Yunxia Gong and colleagues present a review article on aspects of transformation and gene editing techniques in Monascus, focusing on Agrobacterium-mediated transformation, CRISPR and RNAi. The article is comprehensibly written and summarizes relevant information for people working with this fungus and beyond. The references cited are appropriate and the figures are informative.
Comments 1: The article suffers in some passages from poor use of language and needs extensive editing, focusing on grammar, syntax and overall correctness and consistency. It is strongly recommended to seek assistance from a native English speaker.
Answer: Thank you for your kind request. This manuscript has been polished by a professional organization, and relevant certificates are attached.
Comments 2: An additional panel highlighting aspects of Monascus physiology (for example a growth test or a microscopy image) would nicely complement Figure 1.
Figures would really benefit from a more detailed description in the relevant figure legends. In this context, parts of the text describing the principle of each method could be accommodated in the relevant figure legend.
Answer: Thank you for your kind suggestion. Figure 1 has been redrawn, and relevant figure legends have been added for describing figures.
Comments 3: The advantages and/or disadvantages of each method could be further highlighted by adding a few more sentences at the end of each chapter or in the conclusions section.
Answer: Thank you for your kind suggestion. The relevant descriptions of advantages and/or disadvantages for each method have been added and marked in red font, shown on line 104-108, 119-121, 128-135 and 192-195..
Comments 4: The supplementary figures mentioned in the text were not included in the submission and therefore could not be reviewed. Please make sure to include those in your final version.
Round 2
Reviewer 2 Report
The authors took into account most of the comments, removed unnecessary material describing the general principles of some genetic engineering manipulations, added material directly related to Monascus spp. Changed the design of the figures in accordance with the recommendations. The work can be accepted for publication after several corrections related to the changed Figure 1.
Figure 1
What is liophore in fungi. In the resubmitted version of the manuscript you additionally indicated this structure, provide a literary reference to this structure described in Monascus spp. or in representatives of higher taxa.
In the updated version of Figure 1b, the text still runs into the arrows, and the added photographs of the Petri dishes are poorly processed, the Petri dishes are cut off at the side edges, and there is technical noise on the right side of the rightmost Petri dish.
In Figure 1a there is also technical noise, for example after the Aerial hyphae.
Author Response
What is liophore in fungi. In the resubmitted version of the manuscript you additionally indicated this structure, provide a literary reference to this structure described in Monascus spp. or in representatives of higher taxa.
In the updated version of Figure 1b, the text still runs into the arrows, and the added photographs of the Petri dishes are poorly processed, the Petri dishes are cut off at the side edges, and there is technical noise on the right side of the rightmost Petri dish.
In Figure 1a there is also technical noise, for example after the Aerial hyphae.
Answer: Thank you for your kind advice. We have corrected the content of Figure 1, changed liophore to biophore, adjusted the occluded text, and attached the resubmitted Figure 1.
